# Serum soluble B7-H3 is a prognostic marker for patients with non-muscle-invasive bladder cancer

**Takeshi Azuma**[1,2,3]\*, **Yujiro Sato**[1], **Tatsukuni Ohno**[4], **Miyuki Azuma**[2], **Haruki Kume**[3]

**1** Department of Urology, Tokyo Metropolitan Tama Medical Center, Fuchu, Tokyo, Japan, **2** Department of Molecular Immunology, Graduate School of Medical and Dental Sciences, Tokyo Medical and Dental University, Hongo, Tokyo, Japan, **3** Department of Urology, The University of Tokyo Graduate School of Medicine, Hongo, Tokyo, Japan, **4** Oral Health Science Center, Tokyo Dental College, Chiyoda, Tokyo, Japan

\* tazuma-tky@umin.ac.jp

## Abstract

### Background

B7-H3 is a member of the B7 family of immune-regulatory ligands and is a costimulatory molecule promoting the T cell response in vitro. We herein investigated the clinical utility of serum soluble B7-H3 (sB7-H3) in patients with non-muscle invasive bladder cancer (NMIBC).

### Methods

We analyzed 555 patients in whom NMIBC was diagnosed at Tokyo Metropolitan Tama Medical Center between 2008 and 2013. We measured the serum sB7-H3 (sB7-H3) level using the enzyme-linked immunosorbent assay (ELISA) and evaluated the utility of sB7-H3 as a prognostic biomarker for NMIBC. We used the Cox proportional hazards regression model to assess recurrence-free survival (RFS) and progression-free survival (PFS) with the sB7-H3 level.

### Results

We detected high levels of sB7-H3 in the sera of 47% of patients with NMIBC versus only 8% in healthy donors. The increase of sB7-H3 was significantly associated with poor RFS and PFS. Multivariate analysis showed that elevated sB7-H3 was an independent prognostic factor of RFS and PFS. According to the European Organization for Research and Treatment of Cancer (EORTC), in intermediate-low and intermediate-high risk groups, the presence of sB7-H3 significantly determined the rate of recurrence and progression.

### Conclusions

Our data suggested that evaluating serum sB7-H3 expression is a useful tool for predicting the prognosis of patients with NMIBC.

**Data Availability Statement:** All relevant data are within the paper and its Supporting Information files.

**Funding:** This study was supported by JSPS KAKENHI, Grant Number 16K11037 to Y. Sato and JSPS KAKENHI, Grant Number 17K11169 to T. Azuma. The funder had no role in study design, data collection and analysis, decision to publish, or preparation of the manuscript.

**Competing interests:** The authors have declared that no competing interests exist.

## Introduction

Transurethral resection of bladder tumor (TURBT) is usually performed to resect non-muscle invasive bladder cancer (NMIBC) completely [1, 2]. Nonetheless, local recurrence and progression often become a problem despite this procedure. The probability of recurrence at one year is 15–70% while the probability of progression to muscle invasive bladder cancer at five years is 7–40%[3, 4].

Several, clinical, prognostic features are associated with recurrence and progression. Based on these features, the European Association of Urology (EAU) recommends the European Organization for Research and Treatment of Cancer (EORTC) guidelines' classification of patients with NMIBC into four groups to predict recurrence and progression [5, 6]. Although this classification system is helpful for predicting the recurrence and progression of NMIBC, it fails to take into consideration a number of newly reported factors.

Several recent studies have reported an association between inflammation and cancer [7, 8]. In the tumor microenvironment, activation of systemic inflammation, such as the release of several inflammatory cytokines and migration of leucocytes, was observed. The inflammatory cells themselves and the cytokines released by them cause a vicious circle in the tumor microenvironment, where the inflammatory reaction conduces to the growth and progression of the cancer, thus affecting the prognosis of the patient [9, 10]. Recently, we demonstrated that serum soluble B7-H4 could be used as a tool for predicting the prognosis of patients with renal cell carcinoma. Moreover, Podojil reported that B7-H4 can be a new target for immunotherapy in an N-butyl-N-(4-hydroxybutyl)-nitrosamine-induced, murine bladder cancer model [11].

B7-H3 (CD276), which was identified as a new costimulatory molecule activating the T cell response [12], has a single IgV- and IgC-like domain (2Ig form) with a transmembrane and intracellular tail. In humans, a duplicate of B7-H3 (4Ig form) was also identified, but the physiological differences between the 2Ig and 4Ig forms have not yet been elucidated [13, 14]. In addition to the membrane form of B7-H3, serum soluble B7-H3 (sB7-H3), which is involved in the regulation of immune responses, has also been identified. B7-H3 is expressed in various types of cancer, suggesting that it may be associated with inhibition against the anti-tumor immune response [15]. Therefore, in the present study we evaluated the utility of sB7-H3 as a prognostic marker for NMIBC recurrence and progression.

## Materials and methods

### Patients and healthy donors

We studied serum samples from 555 patients with the diagnosis of NMIBC based on histopathological evaluation at Tokyo Metropolitan Tama Medical Center between 2008 and 2013. All the patients were treated with TURBT and were subsequently followed-up. The serum samples were obtained before TURBT and frozen at -80°C. The control group included healthy donors (HD) with no history of malignancy. The study was approved by the ethical review board of Tama Medical Center and was conducted in accordance with the principles of the Declaration of Helsinki and Good Clinical Practice Guidelines. The patients and healthy volunteers provided written informed consent for the specimens to be collected. All the participants provided their informed consent.

### Detection of sB7-H3

To detect human sB7-H3 (human CD276/B7-H3), sandwich enzyme-linked immunosorbent assay (ELISA) (LifeSpan BioScience, Seattle, United States) was performed according to the

manufacturer's instructions. The samples were positive for above-background levels of sB7-H3. Each sample was assayed in triplicate.

## Statistical analysis

The Mann-Whitney U test was used to analyze the serum sB7-H3 level in the NMIBC patients and HD. The $\chi^2$ test or Fisher's exact test was used to analyze the relationship between the presence of B7-H3 and NMIBC characteristics.

Cystoscopy, urine analysis, and cytology were performed every three months in the follow-up examinations after TURBT. We defined recurrence as the first bladder tumor relapse confirmed by pathological examination and the time to recurrence as the time from the date of TURBT to the date of bladder cancer recurrence. We defined progression as the first bladder tumor relapse beyond stage T1 confirmed by pathological examination and the time to progression to MIBC as the time from the date of TURBT to the date of the first progression. We estimated the recurrence and progression curves using the Kaplan-Meier method and used the log-rank test to compare the curves. To evaluate the effect of multiple independent prognostic factors on the outcome, we used Cox regression analysis. We used the JMP® statistical software package for all analyses. $P < 0.05$ indicated statistical significance.

## Results

### Patient characteristics

Table 1 summarizes the characteristics of patients with bladder cancer. We enrolled 555 patients with the diagnosis of NMIBC (445 male and 110 female patients; mean age = 73.5 y; age range: 22–98 y) and 108 HD (81 male and 22 female subjects; mean age = 71.5 y; age range: 35–85 y) as controls and obtained serum samples. Of these, 25 (4%) cases were pTis, 354 (64%) were pTa, and 176 (32%) were pT1. There were 328 (59%) cases of primary bladder cancer and 227 (41%) cases of recurrent bladder cancer. The median follow-up time was 18.2 months.

Three hundred fifteen (57%) patients had a recurrence. We followed up survivors with a recurrence for a median period of 17.9 months. The recurrence-free survival rate (RFS) at one and three years was 44.4% and 38.6%, respectively. The median time of recurrence was 9.4 months. Progression to MIBC occurred in 47 (14.9%) of the 315 patients with recurrence. The progression-free survival rate (PFS) at three and five years was 91.8 and 85.4%, respectively.

### Presence of serum sB7-H3 in patients with NMIBC

ELISA revealed that 47% (261 out of 555) of the serum samples from patients with bladder cancer and 8% (eight of 103) of samples from HD were positive for **sB7-H3**. The levels of sB7-H3 were significantly higher in patients with bladder cancer (31.6 ng/mL in all patients with bladder cancer and 67.1 ng/ml in positive patients with bladder cancer; range: 10–215 ng/mL) than in the HD (1 ng/mL in all HD and 12.5 ng/ml in positive HD; range: 0–17 ng/mL) ($p < 0.001$; Fig 1). Receiver operating curve (ROC) analysis showed that a cut-off value 19.0 ng/ml (sensitivity: 0.45, specificity: 1.0) was able to distinguish bladder cancer patients from the healthy donors (AUC = 0.71, $p < 0.05$). Table 2 shows that sB7-H3-positivity in patients with NMIBC correlated with tumor number and size.

Forty-five percent (148 of 328) of the samples were obtained from the patients with primary bladder cancer and 49% (113 of 227) were obtained from the patients with recurrent bladder cancer. There was no significant difference in serum sB7-H3 level between the patients with primary bladder cancer (average: 28.7 ng/mL; range: 0–215 ng/mL) and those with recurrent bladder cancer (average: 35.7 ng/mL; range: 0–211 ng/mL) (p = 0.07).

Table 1. Characteristics of patients with NMIBC.

| Age | Average | 73.5 years |
|---|---|---|
| | Range | 22-98 years |
| | | Number of patients (%) |
| Gender | Male | 445 (80) |
| | Female | 110 (20) |
| | | |
| Soluble B7-H3 | Negative | 294 (53) |
| | Positive | 261 (47) |
| | | |
| Number of tumors | 1 | 279 (50) |
| | 2-7 | 235 (42) |
| | $\geq 8$ | 41 (8) |
| | | |
| Tumor size | <3 | 497 (90) |
| | $\geq 3$ | 58 (10) |
| | | |
| Prior recurrence rate | Primary | 328 (59) |
| | $\leq 1$ recurrence/year | 200 (36) |
| | >1 recurrence/year | 27 (5) |
| | | |
| T category | Ta | 354 (64) |
| | T1 | 201 (36) |
| | | |
| Carcinoma in situ | No | 460 (83) |
| | Yes | 95 (17) |
| | | |
| Grade | G1 | 80 (14) |
| | G2 | 330 (59) |
| | G3 | 145 (27) |

One hundred forty-three (26%) patients had intravesical Bacillus Calmette-Guérin (BCG) instillation. There was no significant difference in serum sB7-H3 level between patients with a BCG instillation (average: 31.6 ng/mL) and those without it (average: 31.5 ng/mL) (p = 0.98).

## Association between the presence of sB7-H3 and clinical outcomes

The serum sB7-H3-positive and negative groups differed significantly in terms of their recurrence rate (Fig 2A). The RFS rate for the serum sB7-H3-positive and negative groups was 25.4 and 60.2% at three years and 23.2 and 51.9% five years, respectively. Univariate Cox regression analysis revealed that serum sB7-H3 presence was a significant factor influencing bladder tumor recurrence (Table 3). The previously reported factors, including the number of tumors, tumor size, pathological T stage, presence of a carcinoma in situ, and tumor grade, were also associated with a poor clinical outcome on univariate regression analysis (Table 3).

The serum sB7-H3-positive and negative groups differed significantly in terms of their PFS (Fig 2B). PFS for the serum sB7-H3-positive and negative groups was 85.0 and 95.0% at three years and 68.8 and 91.7% at five years, respectively. Univariate Cox regression analysis revealed that the presence of serum sB7-H3 significantly influenced bladder tumor progression (Table 4). The previously reported factors, including the tumor size, pathological T stage,

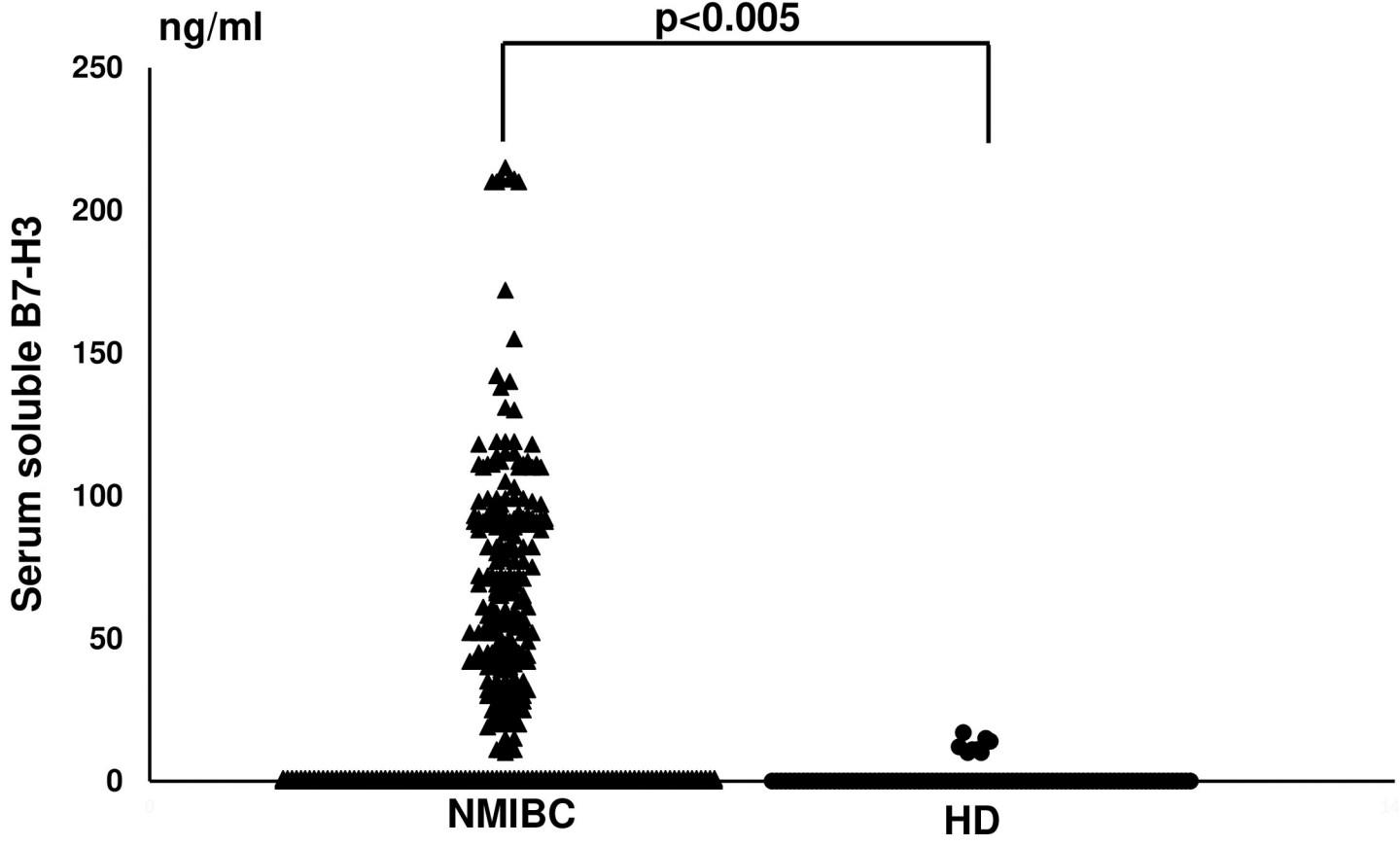

**Fig 1. Detection of serum soluble B7-H3 (sB7-H3) in patients with non-muscle invasive bladder cancer (NMIBC).** We next examined the concentration of sB7-H3 in the HD and patients with NMIBC. Sera from 555 patients with NMIBC and 103 HD were diluted to 1:10 in PBS and tested by ELISA as described in Materials and Methods. The data were analyzed using the Mann-Whitney U test followed by multiple regression analysis (p<0.005).

presence of carcinoma in situ, and tumor grade, were also associated with a poor clinical outcome on univariate regression (Table 4).

Multivariate Cox proportional hazards regression including the presence of serum sB7-H3, number of tumors, tumor size, T category, presence of an in situ carcinoma, grade, and prior recurrence rate revealed that the presence of serum sB7-H3 had a significant, independent, predictive value for RFS and PFS (Tables 3 and 4).

### Association between sB7-H3 presence and the EORTC risk table

We applied the EORTC risk scoring system to our cohort, classifying our patients into low (n = 36; 6%), intermediate-low (n = 239; 43%), intermediate-high (n = 254; 46%), and high risk (n = 26; 5%)-of-recurrence groups (Table 5). We also divided them into low (n = 96; 17%), intermediate-low (n = 204; 37%), intermediate-high (n = 196; 35%), and high risk (n = 59; 11%)-of-progression groups. We further subdivided each group based on the presence of sB7-H3 (Table 5), then compared the RFS and PFS rates against the presence of sB7-H3. In the intermediate-low and high groups, the presence of sB7-H3 was associated with a significantly poorer clinical outcome (Figs 3 and 4). The three-year recurrence rate in patients with sB7-H3 versus those without it was lower in the low (35.7 vs 64.6%, respectively), intermediate-low (22.9 vs 56.9%, respectively), intermediate-high (21.5 vs 44.5%, respectively), and high risk (14.7 vs 44.4%, respectively)-of-recurrence groups. Five-year PFS was also lower in patients

**Table 2. Characteristics of patients with and without soluble B7-H3.**

| | | Patients without soluble B7-H3 | Patients with soluble B7-H3 | p value |
|---|---|---|---|---|
| Number of tumors | 1 | 162 (55) | 117 (45) | |
| | 2-7 | 111 (38) | 124 (48) | |
| | ≥8 | 21 (7) | 20 (7) | <0.05 |
| | | | | |
| Tumor size | <3cm | 270(92) | 227 (87) | |
| | >3cm | 24 (8) | 34 (13) | <0.05 |
| | | | | |
| Prior recurrence rate | Primary | 178 (61) | 150 (57) | |
| | ≤1 recurrence/year | 100 (34) | 100 (38) | |
| | >1 recurrence/year | 16 (5) | 11 (5) | 0.136 |
| | | | | |
| T category | Ta | 191 (65) | 163 (62) | |
| | T1 | 103 (35) | 98 (38) | 0.23 |
| | | | | |
| Carcinoma in situ | No | 233 (79) | 225 (86) | |
| | Yes | 61 (21) | 36 (14) | <0.05 |
| | | | | |
| Grade | G1 | 49 (17) | 31 (12) | |
| | G2 | 169 (57) | 161 (62) | |
| | G3 | 76 (26) | 69 (26) | 0.31 |
| | | | | |
| Recurrence | Yes | 135 (46) | 180 (69) | |
| | No | 159 (54) | 81 (31) | <0.05 |
| | | | | |
| Progression | Yes | 24 (8) | 23 (9) | |
| | No | 270 (92) | 238 (91) | 0.52 |

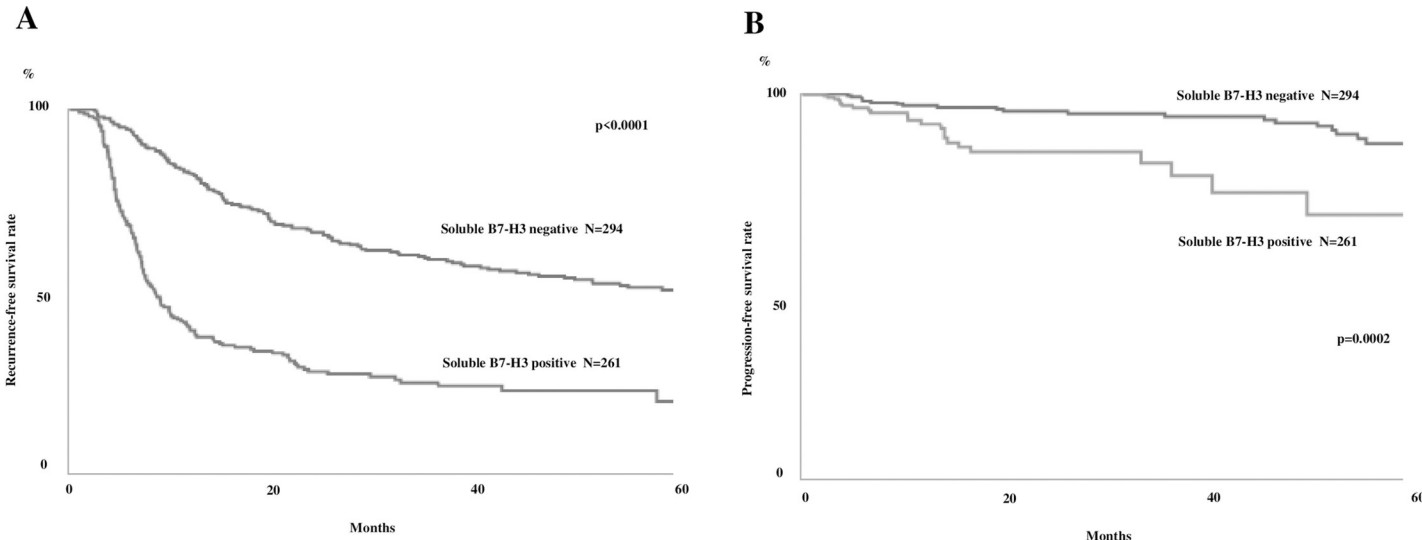

**Fig 2.** (A) Kaplan-Meier recurrence-free survival rate curves for patients with non-muscle invasive bladder cancer (NMIBC) with (n = 261) and without (n = 294) serum soluble B7-H3 (sB7-H3) (B) Progression-free survival rate curves for NMIBC patients with (n = 261) and without (n = 294) serum soluble sB7-H3.

**Table 3. Univariate and multivariate analysis of recurrence.**

| | | Univariate | | Multivariate | |
|---|---|---|---|---|---|
| | Cutoff | HR (95% CI) | p value | HR (95% CI) | p value |
| Soluble B7-H3 | Negative vs Positive | 0.54 (0.43-0.66) | <0.0001 | 0.51 (0.40-0.63) | <0.0001 |
| Number of tumors | | | <0.0001 | | <0.0001 |
| | Single vs 2-7 | 0.43 (0.24-0.60) | | 0.41 (0.23-0.59) | |
| | 2-7 vs >7 | 0.20 (0.031-0.38) | | 0.20 (0.026-0.38) | |
| Tumor size | ≤3 vs >3 cm | 0.30 (0.12-0.46) | 0.0005 | 0.27 (0.086-0.45) | 0.0051 |
| T category | Ta vs T1 | 0.12 (0.0044-0.23) | 0.039 | 0.023 (-0.15-0.11) | 0.73 |
| Carcinoma in situ | No vs Yes | 0.17 (0.024-0.34) | 0.027 | 0.20 (0.049-0.37) | 0.0094 |
| Grade | | | 0.0039 | | 0.011 |
| | G1 vs G2 | 0.40 (0.17-0.65) | | 0.36 (0.12-0.63) | |
| | G2 vs G3 | 0.18 (0.023-0.35) | | 0.18 (0.015-0.35) | |
| Prior recurrence rate | | | 0.079 | | 0.05 |
| | Primary vs ≤1rec/yr | 0.22 (0.015-0.46) | | 0.27 (0.050-0.51) | |
| | ≤1rec/yr vs >1rec/yr | 0.15 (-0.076-0.39) | | 0.085 (-0.14-0.33) | |

with sB7-H3 than in patients without sB7-H3 in the intermediate-low (91.2 vs 98.5%, respectively), intermediate-high (46.4 vs 81.6%, respectively), and low (71.1 vs 81.3%, respectively)-risk-of-progression groups. The low-risk groups had no patients with progression.

## Discussion

In the present study, we revealed that NMIBC with sB7-H3 had a poor clinical prognosis. Especially in groups classified as intermediate-low or high-risk by the EORTC, the presence of sB7-H3 may be a significant predictive factor of recurrence and progression. Groups with intermediate-low and intermediate-high risk had as poor a clinical prognosis as the high-risk group. The ability to detect patients with a poor prognosis in groups with intermediate-low and intermediate-high risk is obviously of great clinical value. About half the patients in the intermediate-low and intermediate-high risk groups benefited from having their serum sB7-H3 level measured.

Several studies reported the expression of B7-H3 in various types of human cancer cell lines and tissues, such as leukemia, colon cancer, lung cancer, melanoma, prostate cancer, urothelial cell cancer, renal cell cancer, gastric cancer, ovarian cancer, and neuroblastoma [16–20]. These data indicate that human cancer cells commonly express B7-H3 on the cell surface, suggesting that B7-H3 plays an important role in cancer cell development.

**Table 4. Univariate and multivariate analysis of progression.**

| | | Univariate | | Multivariate | |
|---|---|---|---|---|---|
| | Cutoff | HR (95% CI) | p value | HR (95% CI) | p value |
| Soluble B7-H3 | Negative vs Positive | 0.56 (0.25-0.87) | 0.0002 | 0.46 (0.14-0.78) | 0.0048 |
| Number of tumors | Single vs Multiple | 0.11 (-0.18-0.42) | 0.089 | 0.75 (-1.2-1.4) | 0.95 |
| Tumor size | ≤3 vs >3 cm | 1.1 (0.76-1.4) | <0.0001 | 0.77 (0.41-1.1) | <0.0001 |
| T category | Ta vs T1 | 12 (5.2-14) | <0.0001 | 12 (8.3-14) | <0.0001 |
| Carcinoma in situ | No vs Yes | 0.62 (0.12-1.3) | 0.026 | 4.5 (0.027-0.67) | 0.033 |
| Grade | G1 and G2 vs G3 | 0.86 (0.57-1.2) | <0.0001 | 1.75 (1.01-3.05) | 0.046 |
| Prior recurrence rate | Primary vs recurrent | 0.15 (-0.15-0.46) | 0.8 | 1.0 (-0.28-1.3) | 0.8 |

**Table 5. Characteristics of patients with and without soluble B7-H3.**

| | | Patients without | Patients with | p value |
|---|---|---|---|---|
| | | Soluble B7-H3 | Soluble B7-H3 | |
| **EORTC recurrence** | **Low** | 22 (7) | 14 (5) | |
| | **Intermediate-low** | 134 (46) | 105 (40) | |
| | **Intermediate-high** | 129 (44) | 125 (48) | **0.58** |
| | **High** | 9 (3) | 17 (7) | |
| | | | | |
| **EORTC progression** | **Low** | 59 (20) | 37 (14) | |
| | **Intermediate-low** | 103 (35) | 104 (40) | **0.42** |
| | **Intermediate-high** | 99 (34) | 95 (36) | |
| | **High** | 33 (11) | 25 (10) | |

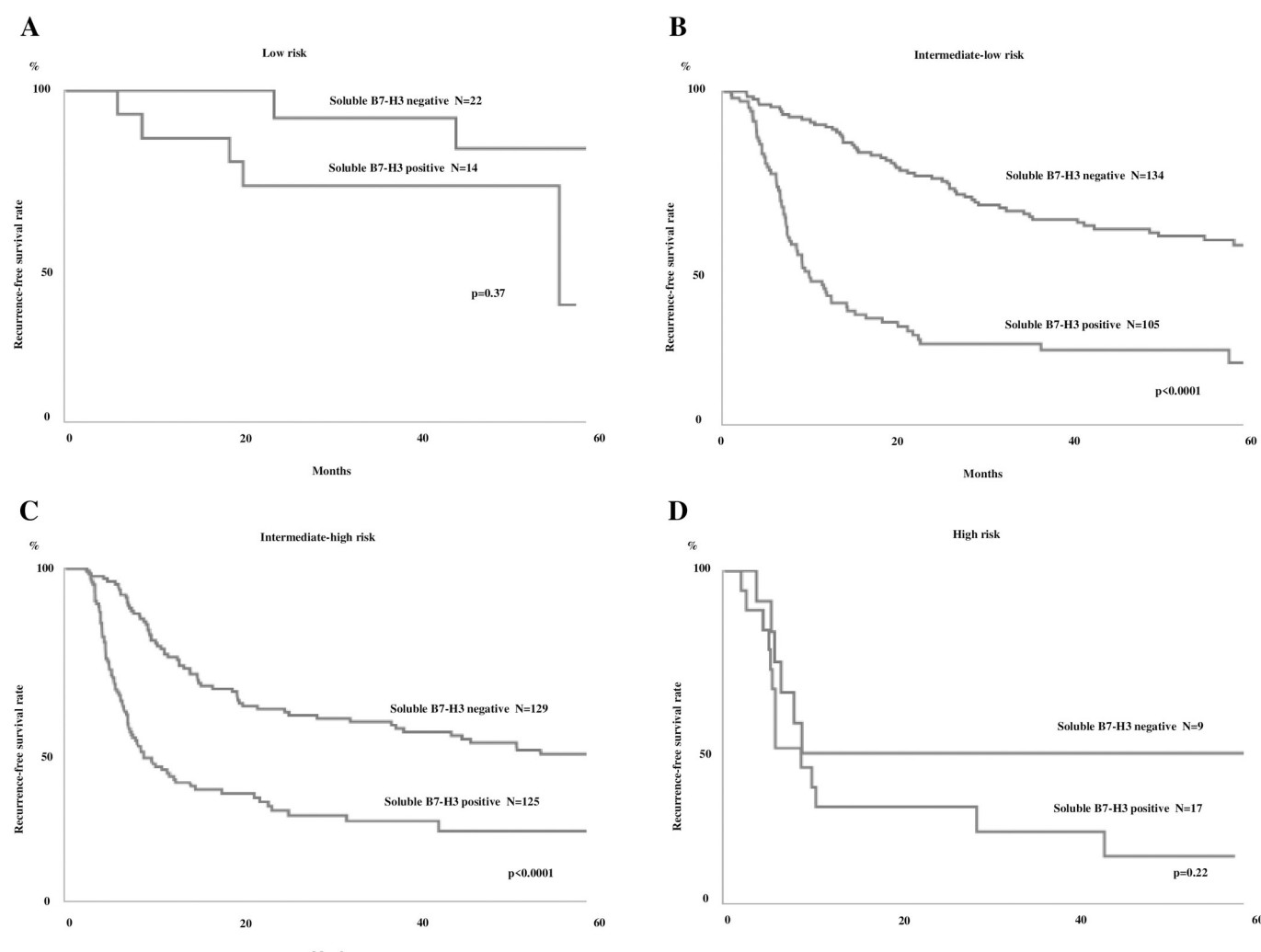

**Fig 3. Kaplan-Meier recurrence-free survival rate curves for patients with non-muscle invasive bladder cancer (NMIBC) classified by the EORTC risk table for recurrence.** All the groups were further divided according to the presence of serum soluble B7-H3. The figures show the low (A), intermediate-low (B), intermediate-high (C), and high (D)-risk groups.

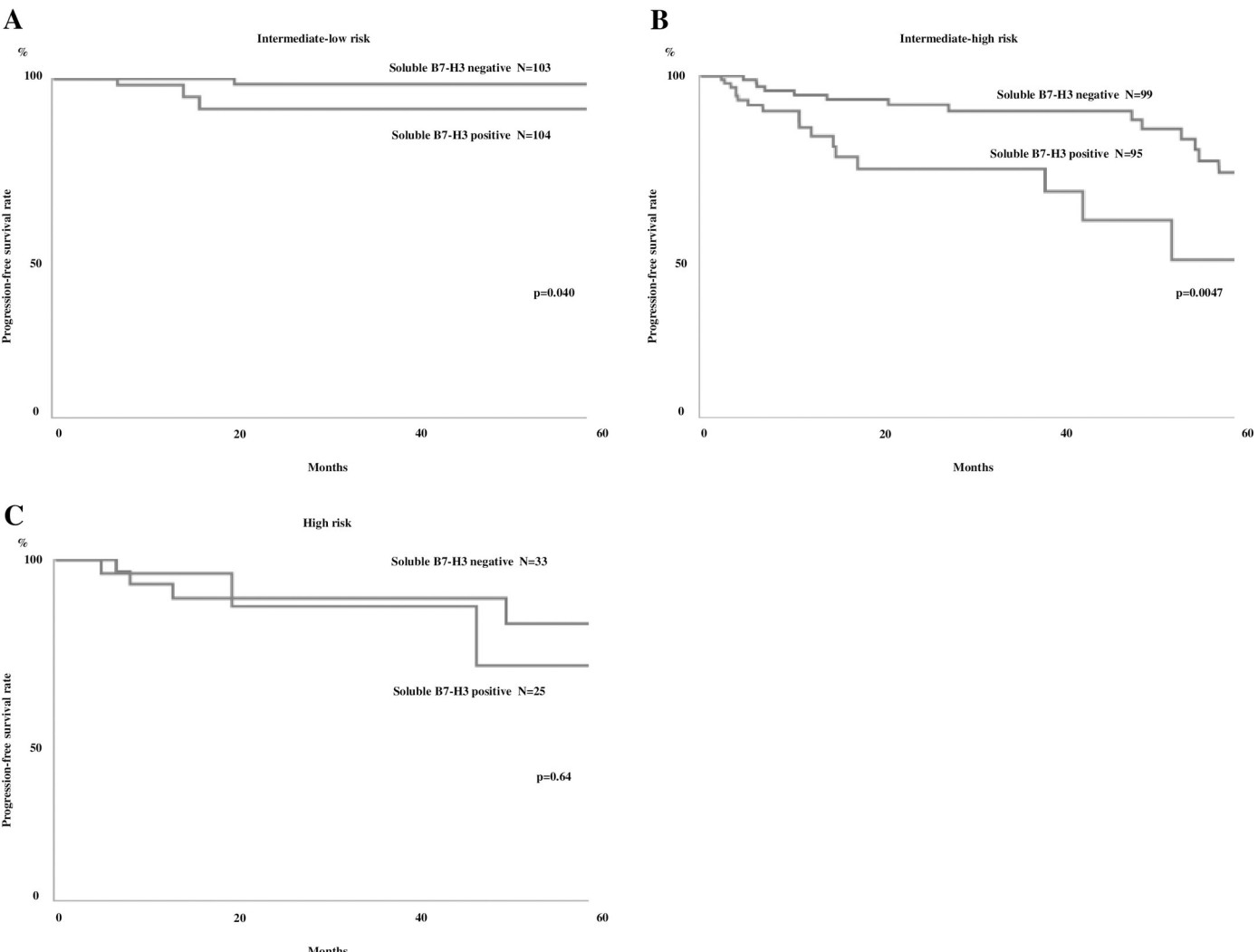

**Fig 4. PFS rate curves for NMIBC patients classified by the EORTC risk table for progression all the groups were further divided according to the presence of sB7-H3.** Figures show the intermediate-low (A), intermediate-high (B) and high (C)-risk groups. The low-risk groups had no patients with progression.

Several studies demonstrated that B7-H3 co-stimulates the T cell response [12]. The co-stimulatory effect of B7-H3 on the CD8+ T-cell response was observed by inducing cytotoxic lymphocytes (CTLs) and anti-tumor immunity in B7-H3-transfected tumor cells [15]. In some murine models, B7-H3 expression led to the activation of tumor-specific CTL, which can slow tumor growth or eradicate tumors [15, 21, 22]. These data indicated that a high level of B7-H3 expression may be beneficial in the immune response to tumor antigens. Manipulating B7-H3 may provide an opportunity to regulate the immune response and design new immunotherapeutic approaches.

B7-H3 plays an important role in tumor progression, metastasis, recurrence, and other adverse clinical features [17, 18, 23–25]. Three previous studies demonstrated an association between B7-H3 expression and cancer-specific mortality or disease progression after a cystectomy [26–28]; two of the studies showed no association while one did. The discrepancies in the findings may have been due to a higher proportion of muscle-invasive bladder cancer (85.8%) in Zhili's cohort than in Boorjian et al.'s cohort (65.2%). These studies focused on mB7-H3

and targeted populations with a radical cystectomy, i.e., aggressive cancer. In the present study, we focused on sB7-H3 in patients with NMIBC and demonstrated an association between the presence of sB7-H3 and the recurrence and progression rates. We found that the presence of sB7-H3 may be an independent factor predictive of a good prognosis in terms of recurrence and progression in patients with NMIBC.

sB7-H3 is a soluble form of B7-H3 that is produced by monocytes, activated T cells, DCs, and B7-H3-positive tumor cells [29]. Zhang et al. showed that patients with the diagnosis of sepsis, in contrast to healthy individuals, exhibited significantly higher levels of plasma sB7-H3 (sB7-H3), which correlated with the clinical outcomes and levels of plasma TNF-α and IL-6 [30]. They also reported that membrane-bound sB7-H3 was cleaved from leukocytes by matrix metalloproteinases [29].

sB7-H3 were able to be detected in several cancers and cancer cell lines, and high serum levels were significantly associated with poor clinical outcomes [31, 32]. Chao et al. demonstrated the invasion and metastasis of pancreatic cancer cells via the TLR4/NF-κB signaling pathway [33]. In addition, sB7-H3 induced VEGF and IL-8 secretion via the TLR4/NF-κB pathway, supporting the idea that sB7-H3 promotes invasion and cancer cell metastasis. Hao et al. showed that mB7-H3 can regulate epithelial to mesenchymal transition of hepatocellular carcinoma through activating Jak2/Stat3 signaling [34]. According to these two reports, mB7-H3 or sB7-H3 was associated with inflammation, and their signaling pathways were different.

TLR4 signaling in cancer cells and leukocytes mediate inflammation related to oncogenesis [7, 35]. Immune cells infiltration occurs in various cancers, with interaction between tumor cells and lymphocytes forming an inflammatory positive feedback loop [36]. TLR4, which is expressed in cancer cells and leukocytes, mediates inflammation, which often leads to the suppression of immune functions, inducing tumor immune escape [37]. Several studies have reported that the growth and invasion of cancer cells may be able to be boosted by inflammation in the tumor microenvironment [38]. Zhang et al. showed that sB7-H3 was able to augment the inflammatory response thorough TLR4 [30]. Our results suggested that serum sB7-H3 amplified cancer-associated inflammation in bladder cancer patients.

## Conclusions

NMIBC with sB7-H3 has a poor clinical prognosis. The presence of sB7-H3 may be useful as a predictive factor for recurrence and progression. These findings may be helpful for determining an appropriate treatment for patients with NMIBC.

## Supporting information

**S1 Data.**
(XLSX)

## Author Contributions

**Conceptualization:** Takeshi Azuma, Miyuki Azuma.

**Data curation:** Yujiro Sato.

**Funding acquisition:** Takeshi Azuma, Yujiro Sato.

**Investigation:** Takeshi Azuma.

**Methodology:** Takeshi Azuma, Tatsukuni Ohno.

**Supervision:** Tatsukuni Ohno, Miyuki Azuma, Haruki Kume.

**Writing – original draft:** Takeshi Azuma.

**Writing – review & editing:** Takeshi Azuma, Miyuki Azuma, Haruki Kume.

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
