## [Decision Letter · Decision Letter 0]

23 Sep 2020

PONE-D-20-26604

Serum soluble B7-H3 is a prognostic marker for patients with non-muscle-invasive bladder cancer.

PLOS ONE

Dear Dr. Azuma,

Thank you for submitting your manuscript to PLOS ONE. After careful consideration, we feel that it has merit but does not fully meet PLOS ONE’s publication criteria as it currently stands. Therefore, we invite you to submit a revised version of the manuscript that addresses the points raised during the review process.

Please submit your revised manuscript within 3 months. If you will need more time than this to complete your revisions, please reply to this message or contact the journal office at plosone@plos.org. Please include the following items when submitting your revised manuscript:

We look forward to receiving your revised manuscript.

Kind regards,

Francisco X. Real

Academic Editor

PLOS ONE

Journal Requirements:

https://journals.plos.org/plosone/article?id=10.1371/journal.pone.0199719 (Introduction paragraph 3)

https://www.jimmunol.org/content/185/6/3677.full (Discussion paragraph 5)

https://www.nature.com/articles/srep27528 (Discussion paragraph 7)

https://www.clinical-genitourinary-cancer.com/article/S1558-7673(13)00049-9/fulltext (Discussion paragraph 7)

In your revision ensure you cite all your sources (including your own works), and quote or rephrase any duplicated text outside the methods section. Further consideration is dependent on these concerns being addressed.

3. Please provide a brief summary of the methods regarding the detection and measurement of sB7-H3 in serum. This is in line with our reproducibility criterion for publishing, see https://journals.plos.org/plosone/s/criteria-for-publication.

4. In your Methods section, please provide additional information about the collection of control tissue specimens used in this study, the method used to collect them, and the demographic details of the patients from which they were collected. Please ensure you have provided sufficient details to replicate the analyses such as: a) the  date range (month and year) during which you collected specimens,  b) a description of how participants were recruited to provide samples, and c) eligibility criteria for being included in this part of the study.

Reviewers' comments:

Reviewer's Responses to Questions

**Comments to the Author**

1. Is the manuscript technically sound, and do the data support the conclusions?

Reviewer #1: Yes

Reviewer #2: Partly

Reviewer #3: Yes

2. Has the statistical analysis been performed appropriately and rigorously? 

Reviewer #1: Yes

Reviewer #2: Yes

Reviewer #3: No

3. Have the authors made all data underlying the findings in their manuscript fully available?

Reviewer #1: Yes

Reviewer #2: Yes

Reviewer #3: Yes

4. Is the manuscript presented in an intelligible fashion and written in standard English?

Reviewer #1: No

Reviewer #2: Yes

Reviewer #3: Yes

5. Review Comments to the Author

Reviewer #1: This manuscript is addressing a very interesting question asking if B7-H4, a specific member of the immunstimulatory B7 family, in its soluble from can be used as a prognostic marker in NMIBC.

A strength of this project is its large patient cohort.

However, there are several aspects in the manuscript that should be improved.

1) the introduction is not written in a precise and clear style and order.

2) The authors should carefully ensure that they really describe what they are doing. (for example under the headline "presence of serum sB7-H3 in patients with NMIBC" they start with "This assay revealed that..." without mentioning which assay they mean).

3) There is a very recent paper from Podojil et al., on B7-H4 in urothelial carcinoma that should be included into the discussion ( PMID: 32363111 ).

Reviewer #2: This is a review of a historical series of 555 patients with NMIBC, in which sB7-H3 has been measured in plasma, relating it to progression and recurrence-free survival.

The authors should be congratulated for an exhaustive work, as well as the description of a promising marker in the study of recurrence and progression of NMIBC. Anyway, I think the work may improve with the following contributions:

Introduction: The first paragraph seems to be missing. If not, it is advisable to check the grammar.

Material and methods: Given that the primary objective is recurrence, it should be indicated how these patients have been followed (cystoscopy, cytology, ultrasound, CT ...). In the definitions part, progression is defined as “beyond T2 stage”. I suppose they mean "beyond T1 stage", to diagnose progression to muscle infiltrating.

It does not show how these patients were treated. Given that most are intermediate or high risk tumours, the logical thing is that they had some treatment. The therapy applied logically may have influenced the recurrence, altering the results of the study. If this data is not available, it should be included among the weaknesses of the work, in the discussion section.

Results: The median follow-up of the entire series is missing, not only of the tumors that have recurred, as well as whether there were any lost to follow-up.

At the bottom of Figure 2, the results of the curves are discussed. This data is already given in the text, so duplication should be avoided.

As there are some healthy donors positive for sB7H3, can a cut-off level be defined to know when it is considered positive?

Given that tumors with positive sB7H3 are usually recurrent, I recommend doing uni / multivariate analysis of the variable “prior recurrence”, grouping primary vs recurrent. It is possible that it may be significant.

Discussion

The analysis has been carried out on a series for which the treatment applied is not available. If the percentage of patients who have received BCG is high, the EORTC tables loses validity, making it more advisable to use the risk according to CUETO (Babjuck et al, EAU Guidelines NMIBC 2020). If the information is available, it is advisable to see if these results hold up. If so, the importance of this marker would increase, as it would be significant even when a more “modern” BCG treatment is performed. And it can serve as an indicator to predict response to BCG, something much needed today.

Conclusions

Nothing to add

Reviewer #3: The manuscript “Serum soluble B7-H3 is a prognostic marker for patients with non-muscle-invasive bladder cancer“ by et al address an important point in NMIBC that is the finding of good prognostic marker for recurrence and progression of the disease. The has evaluated 555 patient samples and found a correlation between the serum levels of B7-H3 and disease recurrence and progression to MIBC. The article is well written but I miss some aspect that I think need to be elucidated.

1.- The authors point the NMIBC need of good markers for disease progression, but they not make a reasoning of why to measure B7-H3, is there any rationale for that or is just because?

2.- With the patient descriptions it is not clear how many of the patients are primary or recurrent in each analysis. It will be interesting to have the B7-H3 values dissected by patients’ groups not just by positive vs negative. Is there a cut value that really distinguish healthy vs cancer patient?

3.-Currently the patients that are considered high risk of tumor progression according to EORT guidelines are treated with intravesical instillations. Is that the case for some of those patients in the cohort?. In the same line, it will be interesting to know how are the B7-H3 serum levels in a cystitis or a bladder infection patient. Will B7-H3 be as HD or as a cancer patient?

4.- I miss the analysis of how varied (or not) the B7-H3 serum levels among the patients with a primary tumor vs those that are recurrent, not only in positive vs negative but the quantitative amount.

5.- In table 2 which is the unit of the tumor size? Cm?. The authors distinguish Ta T1 and Tis, how many of the patients have Tis plus Ta or T1 and how do this affect the analysis?

6.- The authors point in mat. and meth. that the samples were collected between 2008 and 2013. I thnk it will be important to know the follow up time.

7.- In page 16, last paragraph, the author wrote that in the samples they found an 8% of tumor progression (47 samples) but in figure 4 they make the Kaplan analysis with the 555 samples. I think they should make this among the samples of recurrent patients not with the total.

8.- And last, I think that the authors most do an effort to point clearly how many of the patients that are not categorized now of high risk of recurrence or progression will be benefit with the measurement of sB7-H3. To me it is not clear as it is now.

6. PLOS authors have the option to publish the peer review history of their article (what does this mean?). If published, this will include your full peer review and any attached files.

Reviewer #1: No

Reviewer #2: No

Reviewer #3: No

---

## [Author Response · Author response to Decision Letter 0]

28 Oct 2020

Reviewer #1: This manuscript is addressing a very interesting question asking if B7-H4, a specific member of the immunstimulatory B7 family, in its soluble from can be used as a prognostic marker in NMIBC.

A strength of this project is its large patient cohort.

However, there are several aspects in the manuscript that should be improved.

1) the introduction is not written in a precise and clear style and order.

Thank you for your suggestion. Accordingly, we have written, “Transurethral resection of bladder tumor (TURBT) is usually performed to resect non-muscle invasive bladder cancer (NMIBC) completely” in the Introduction.

2) The authors should carefully ensure that they really describe what they are doing. (for example under the headline "presence of serum sB7-H3 in patients with NMIBC" they start with "This assay revealed that..." without mentioning which assay they mean).

Thank you for your suggestion. Accordingly, we have written, “ELISA revealed that 47% (261 out of 555) of the samples…”

3) There is a very recent paper from Podojil et al., on B7-H4 in urothelial carcinoma that should be included into the discussion ( PMID: 32363111 ).

Thank you for your suggestion. Accordingly, we have added “Podojil reported that B7-H4 can be a new target for immunotherapy in an N-butyl-N-(4-hydroxybutyl)-nitrosamine-induced, murine bladder cancer model” in the Introduction.

Reviewer #2: This is a review of a historical series of 555 patients with NMIBC, in which sB7-H3 has been measured in plasma, relating it to progression and recurrence-free survival.

The authors should be congratulated for an exhaustive work, as well as the description of a promising marker in the study of recurrence and progression of NMIBC. Anyway, I think the work may improve with the following contributions:

Introduction: The first paragraph seems to be missing. If not, it is advisable to check the grammar.

Thank you for your suggestion. Accordingly, we have changed this to “Transurethral resection of bladder tumor (TURBT) is usually performed to resect non-muscle invasive bladder cancer (NMIBC) completely” in the Introduction.

Material and methods: Given that the primary objective is recurrence, it should be indicated how these patients have been followed (cystoscopy, cytology, ultrasound, CT ...). In the definitions part, progression is defined as “beyond T2 stage”. I suppose they mean "beyond T1 stage", to diagnose progression to muscle infiltrating.

Thank you for your suggestion. Accordingly, we added, “Cystoscopy, urine analysis, and cytology were performed every three months in the follow-up examinations after TURBT” in Materials and Methods. We have also corrected to “beyond stage T1.”

It does not show how these patients were treated. Given that most are intermediate or high risk tumours, the logical thing is that they had some treatment. The therapy applied logically may have influenced the recurrence, altering the results of the study. If this data is not available, it should be included among the weaknesses of the work, in the discussion section.

Thank you for your suggestion. Accordingly, we added “There was no significant difference in serum sB7-H3 level between patients with a BCG instillation (average: 31.6 ng/mL) and those without it (average: 31.5 ng/mL) (p=0.98)” in the Results.

Results: The median follow-up of the entire series is missing, not only of the tumors that have recurred, as well as whether there were any lost to follow-up.

Thank you for your suggestion. Accordingly, we have added “The median follow-up time was 18.2 months” in the Results.

At the bottom of Figure 2, the results of the curves are discussed. This data is already given in the text, so duplication should be avoided.

Thank you for your suggestion. Accordingly, deleted this discussions in Fig.2.

As there are some healthy donors positive for sB7H3, can a cut-off level be defined to know when it is considered positive?

Thank you for your suggestion. However, there is a clear cut-off level. We explained that the samples were positive for above-background levels of sB7-H3.

Given that tumors with positive sB7H3 are usually recurrent, I recommend doing uni / multivariate analysis of the variable “prior recurrence”, grouping primary vs recurrent. It is possible that it may be significant.

Thank you for your suggestion. Accordingly, we added, “Forty-five percent (148 of 328) of the samples were obtained from the patients with primary bladder cancer and 49% (113 of 227) were obtained from patients with recurrent bladder cancer. There was no significant difference in serum sB7-H3 level between patients with primary bladder cancer (average: 28.7 ng/mL; range: 0 - 215 ng/mL) and patients with recurrent bladder cancer (average: 35.7 ng/mL; range: 0 - 211 ng/mL) (p=0.07)” in the Results.

Discussion

The analysis has been carried out on a series for which the treatment applied is not available. If the percentage of patients who have received BCG is high, the EORTC tables loses validity, making it more advisable to use the risk according to CUETO (Babjuck et al, EAU Guidelines NMIBC 2020). If the information is available, it is advisable to see if these results hold up. If so, the importance of this marker would increase, as it would be significant even when a more “modern” BCG treatment is performed. And it can serve as an indicator to predict response to BCG, something much needed today.

Thank you for your suggestion. Accordingly, we added, “There was no significant difference in serum sB7-H3 level between patients with a BCG instillation (average: 31.6 ng/mL) and those without it (average: 31.5 ng/mL) (p=0.98)” in the Results.

Reviewer #3: The manuscript “Serum soluble B7-H3 is a prognostic marker for patients with non-muscle-invasive bladder cancer“ by et al address an important point in NMIBC that is the finding of good prognostic marker for recurrence and progression of the disease. The has evaluated 555 patient samples and found a correlation between the serum levels of B7-H3 and disease recurrence and progression to MIBC. The article is well written but I miss some aspect that I think need to be elucidated.

1.- The authors point the NMIBC need of good markers for disease progression, but they not make a reasoning of why to measure B7-H3, is there any rationale for that or is just because?

Thank you for your suggestion. Accordingly, we added, “B7-H3 is expressed in various types of cancer, suggesting that it may be associated with inhibition against the anti-tumor immune response“ in the Introduction.

2.- With the patient descriptions it is not clear how many of the patients are primary or recurrent in each analysis. It will be interesting to have the B7-H3 values dissected by patients’ groups not just by positive vs negative. Is there a cut value that really distinguish healthy vs cancer patient?

Thank you for your suggestion.　 Accordingly, we added, “Forty-five percent (148 of 328) of the samples were obtained from the patients with primary bladder cancer and 49% (113 of 227) were obtained from patients with recurrent bladder cancer. There was no significant difference in serum sB7-H3 level between patients with primary bladder cancer (average: 28.7 ng/mL; range: 0 - 215 ng/mL) and patients with recurrent bladder cancer (average: 35.7 ng/mL; range: 0 - 211 ng/mL) (p=0.07)” in the Results.

3.-Currently the patients that are considered high risk of tumor progression according to EORT guidelines are treated with intravesical instillations. Is that the case for some of those patients in the cohort?. In the same line, it will be interesting to know how are the B7-H3 serum levels in a cystitis or a bladder infection patient. Will B7-H3 be as HD or as a cancer patient?

Thank you for your suggestion. Accordingly, we added “There was no significant difference in serum sB7-H3 level between patients with a BCG instillation (average: 31.6 ng/mL) and those without it (average: 31.5 ng/mL) (p=0.98)” in the Results. We do not have B7-H3 serum levels for cystitis.

4.- I miss the analysis of how varied (or not) the B7-H3 serum levels among the patients with a primary tumor vs those that are recurrent, not only in positive vs negative but the quantitative amount.

Thank you for your suggestion.　Accordingly, we added “There was no significant difference in serum sB7-H3 level between patients with primary bladder cancer (average: 28.7 ng/mL; range: 0 - 215 ng/mL) and patients with recurrent bladder cancer (average: 35.7 ng/mL; range: 0 - 211 ng/mL) (p=0.07)” in the Results.

5.- In table 2 which is the unit of the tumor size? Cm?. The authors distinguish Ta T1 and Tis, how many of the patients have Tis plus Ta or T1 and how do this affect the analysis?

Thank you for your suggestion. Accordingly, we added “cm” in the Table. 

There were 80 patients with Tis plus Ta or T1. These patients’ distribution was not different from that of patients with pure Tis.

6.- The authors point in mat. and meth. that the samples were collected between 2008 and 2013. I thnk it will be important to know the follow up time.

Thank you for your suggestion. Accordingly, we added “The median follow-up time was 18.2 months” in the Results.

7.- In page 16, last paragraph, the author wrote that in the samples they found an 8% of tumor progression (47 samples) but in figure 4 they make the Kaplan analysis with the 555 samples. I think they should make this among the samples of recurrent patients not with the total.

Thank you for your suggestion. Accordingly, we changed this to “Progression to MIBC occurred in 47 (14.9 %) of the 315 patients with recurrence”.

8.- And last, I think that the authors most do an effort to point clearly how many of the patients that are not categorized now of high risk of recurrence or progression will be benefit with the measurement of sB7-H3. To me it is not clear as it is now.

Thank you for your suggestion. Accordingly, we added “About half the patients in the low-intermediate and high-intermediate risk groups benefited from having their serum sB7-H3 level measured” in Discussion.

---

## [Decision Letter · Decision Letter 1]

20 Nov 2020

Serum soluble B7-H3 is a prognostic marker for patients with non-muscle-invasive bladder cancer.

PONE-D-20-26604R1

Dear Dr. Azuma,

We’re pleased to inform you that your manuscript has been judged scientifically suitable for publication and will be formally accepted for publication once it meets all outstanding technical requirements.

Kind regards,

Francisco X. Real

Academic Editor

PLOS ONE

Additional Editor Comments (optional):

Reviewers' comments:

Reviewer's Responses to Questions

**Comments to the Author**

1. If the authors have adequately addressed your comments raised in a previous round of review and you feel that this manuscript is now acceptable for publication, you may indicate that here to bypass the “Comments to the Author” section, enter your conflict of interest statement in the “Confidential to Editor” section, and submit your "Accept" recommendation.

Reviewer #1: All comments have been addressed

Reviewer #2: All comments have been addressed

Reviewer #3: All comments have been addressed

2. Is the manuscript technically sound, and do the data support the conclusions?

Reviewer #1: (No Response)

Reviewer #2: (No Response)

Reviewer #3: Yes

3. Has the statistical analysis been performed appropriately and rigorously? 

Reviewer #1: (No Response)

Reviewer #2: (No Response)

Reviewer #3: Yes

4. Have the authors made all data underlying the findings in their manuscript fully available?

Reviewer #1: (No Response)

Reviewer #2: (No Response)

Reviewer #3: Yes

5. Is the manuscript presented in an intelligible fashion and written in standard English?

Reviewer #1: (No Response)

Reviewer #2: (No Response)

Reviewer #3: Yes

6. Review Comments to the Author

Reviewer #1: (No Response)

Reviewer #2: (No Response)

Reviewer #3: The authors addressed most of the comments raised the previous review. I think that this manuscript is now acceptable for publication. Congratulations!!

7. PLOS authors have the option to publish the peer review history of their article (what does this mean?). If published, this will include your full peer review and any attached files.

Reviewer #1: No

Reviewer #2: No

Reviewer #3: No

---

## [Editor Report · Acceptance letter]

24 Nov 2020

PONE-D-20-26604R1 

Serum soluble B7-H3 is a prognostic marker for patients with non-muscle-invasive bladder cancer. 

Dear Dr. Azuma:

I'm pleased to inform you that your manuscript has been deemed suitable for publication in PLOS ONE. Congratulations! Your manuscript is now with our production department. 

Kind regards, 

on behalf of

Dr. Francisco X. Real 

Academic Editor

PLOS ONE